# Efficient Path Planning for Mobile Robot Based on Deep Deterministic Policy Gradient

**DOI:** 10.3390/s22093579

**Published:** 2022-05-08

**Authors:** Hui Gong, Peng Wang, Cui Ni, Nuo Cheng

**Affiliations:** 1Information Science and Electrical Engineering, Shandong Jiao Tong University, Jinan 250357, China; gh960120@163.com (H.G.); 205034@sdjtu.edu.cn (C.N.); cn17861403204@163.com (N.C.); 2Institute of Automation, Shandong Academy of Sciences, Jinan 250013, China

**Keywords:** path planning, DDPG, LSTM, reward function, mixed noise

## Abstract

When a traditional Deep Deterministic Policy Gradient (DDPG) algorithm is used in mobile robot path planning, due to the limited observable environment of mobile robots, the training efficiency of the path planning model is low, and the convergence speed is slow. In this paper, Long Short-Term Memory (LSTM) is introduced into the DDPG network, the former and current states of the mobile robot are combined to determine the actions of the robot, and a Batch Norm layer is added after each layer of the Actor network. At the same time, the reward function is optimized to guide the mobile robot to move faster towards the target point. In order to improve the learning efficiency, different normalization methods are used to normalize the distance and angle between the mobile robot and the target point, which are used as the input of the DDPG network model. When the model outputs the next action of the mobile robot, mixed noise composed of Gaussian noise and Ornstein–Uhlenbeck (OU) noise is added. Finally, the simulation environment built by a ROS system and a Gazebo platform is used for experiments. The results show that the proposed algorithm can accelerate the convergence speed of DDPG, improve the generalization ability of the path planning model and improve the efficiency and success rate of mobile robot path planning.

## 1. Introduction

With the rise and continuous development of robot technology, mobile robots are becoming more and more widely used and are playing an important role in more and more fields. Mobile robots can perform tasks in various scenarios, such as package pick-up and delivery in warehouses and guiding patients in hospitals [1,2,3]. As one of the key technologies of mobile robot applications, path planning has become an indispensable part of mobile robots moving towards artificial intelligence. Its purpose is to find an optimal or suboptimal collision-free path from the starting point of the mobile robot’s movement to the target point in the application scenario so as to ensure the rapid and safe movement of the robot and improve the work efficiency. Ideal path planning can greatly save the movement time of mobile robots on the road, help mobile robots complete tasks efficiently and accurately, and provide favorable guarantees for the applications of mobile robots in various industries.

Algorithm design is the core of mobile robot path planning, according to the different application environment and the degree of intelligence, and the current path planning algorithms can be divided into traditional methods and intelligent methods [4,5,6]. The traditional methods mainly include the Dijkstra algorithm, A* algorithm, Artificial Potential Field and Genetic algorithm and so on [7]. The Dijkstra algorithm is a path planning algorithm that was proposed very early. The algorithm takes the starting point as the center origin and spreads layers outward until the shortest paths to all nodes are found [8]. However, the Dijkstra algorithm uses the path length as the weight factor to search for the shortest path, which increases the movement time of the mobile robot and reduces the work efficiency [9]. The A* algorithm is improved on the basis of the Dijkstra algorithm with the addition of heuristic function, which is one of the direct search methods to find the shortest path in a static environment [10]. However, when the A* algorithm is solving the shortest path, it is easily affected by the estimated value of generation, and the algorithm easily falls into the local optimal and has poor stability [11]. In 1994, Khatib and Andrews et al. proposed a virtual force method—the Artificial Potential Field Method—which regarded the movement space of the mobile robot as the potential field in physics. Obstacles generate repulsive force on mobile robots, while target points generate gravitational force on them, and the resultant force of the two is the final movement direction. The algorithm has good real-time performance, a small calculation amount and fast convergence speed, but it is not suitable for complex environments [12,13]. The genetic algorithm was first proposed by John Holland in the 1970s. The algorithm was proposed by simulating the biological evolutionary law in nature and had the characteristics of strong global search ability and good robustness. However, the algorithm takes a long time to encode, and the convergence speed is slow, so it cannot guarantee real-time performance in complex environments [14,15]. Based on the above algorithms, many optimized algorithms have been proposed successively, such as the convex optimization method, Bezier curve method and the optimal-control-theory-based method [16,17,18]. However, most of the traditional path planning methods have problems such as relying on maps and low real-time performance. As the environment in which the mobile robot is located becomes more and more complex, the traditional path planning methods have been unable to adapt to the actual task requirements [19,20].

In recent years, with the rise of artificial intelligence, path planning methods based on deep reinforcement learning have appeared [21,22]. Deep reinforcement learning combines the perception ability of deep learning with the decision-making ability of reinforcement learning. In the process of continuous interaction and trial and error between the mobile robot and the environment, the action strategy is optimized by accumulating rewards. It realizes the learning method from the environment state to behavior mapping, and the computing efficiency is high [23,24]. Q-Learning is a typical reinforcement learning algorithm used for path planning. It uses *Q* value tables to store and update state-action values and updates the *Q* tables according to the empirical knowledge learned by mobile robots. After convergence, the optimal path is obtained according to *Q* value [25]. However, when the environment is complex, there are too many state actions in *Q* tables, which leads to a sharp increase in memory consumption and dimension disaster [26]. In 2013, Google DeepMind combined deep learning with Q-learning, used a neural network to approximate the value function, and proposed deep Q-Learning (DQN) for the first time, which realized an end-to-end revolutionary algorithm from perception to action, and solved the dimension disaster problem in Q-Learning [27]. However, DQN is only applicable to discrete action space, but not to continuous action space [28]. Google DeepMind incorporated DQN into the Actor-Critic framework in 2015 and proposed the Deep Deterministic Policy Gradient (DDPG) to solve the problem of continuous action space. When using DDPG for mobile robot path planning, it can output continuous actions [29]. However, when the environment is complex, the DDPG algorithm easily falls into the local optimum, and there are problems such as a low success rate and slow learning speed [30]. Tai L et al. [31] propose a learning-based mapless motion planner that takes a sparse 10-dimensional range and the position of the target relative to the mobile robot coordinate system as input and continuous steering commands as output, extending DDPG to the asynchronous version to improve sampling efficiency. The results show that the planner can navigate the mobile robot to the desired target location without colliding with any obstacles. However, this method has neither the memory of previous observations nor the ability of long-term prediction, so the planned path is tortuous and not optimal. Jesus J C et al. [32] propose a deep reinforcement learning method that applies DDPG to mobile robot navigation. This method takes the mobile robot to reach the target position in different simulated environments of the task and creates a good reward function. However, the training effect is not very ideal. Peng Li et al. [33] proposed a new DDPG algorithm, which used Rectified Adam (RAdam) to replace the neural network optimizer in DDPG and combined this with curiosity algorithm to improve the training effect, but the convergence speed was not ideal.

Compared with traditional path planning methods, the path planning methods based on deep reinforcement learning do not need to build the whole environment model and can realize self-learning from the environment state to action mapping, which has high flexibility. Through the continuous interaction between the mobile robot and the environment, the deep reinforcement learning uses the corresponding action strategy to determine the next action of the mobile robot according to the state of the robot and combines the reward function to continuously optimize the action strategy. As one of the typical algorithms of deep reinforcement learning applied to path planning, DDPG can train the model in self-constructed simulation environment and be directly applied to the actual environment, with strong generalization ability. However, due to the adoption of deterministic policy, when the robot is in the same state, the actions given are also the same. This will lead to a single action of mobile robots that cannot fully explore the environment and may not be able to reach the target point with the optimal path. Especially in complex environments, due to the limited observable range of mobile robots and the lack of previous “memory” of DDPG, it is unable to collect enough state information to train the algorithm, resulting in low efficiency, slow convergence speed and low success rate in the training of the algorithm.

This paper fully analyzes the advantages and disadvantages of DDPG. Based on reference [31], the efficiency and success rate of DDPG algorithm in path planning are further improved by introducing LSTM, optimal design reward function and mixed noise. The main contributions are as follows: (1) The “memory” ability of LSTM is utilized to optimize the DDPG network structure, and Batch Norm layer is added after each layer of the Actor network to improve network stability and speed up algorithm convergence. (2) By optimizing the reward function, the mobile robot is guided to move faster towards the target point. (3) Different normalization methods are used to normalize the distance and angle between mobile robot and target point to improve the learning efficiency of the path planning model. (4) A mixed noise composed of Gaussian noise and Ornstein–Uhlenbeck (OU) noise is designed to make the learning process of mobile robot have higher randomness, avoid falling into local optimum, and improve the exploration efficiency of mobile robot.

The structure of this paper is as follows, Section 2 describes and analyzes the correlation algorithms. Section 3 introduces the core method of this paper in detail, namely the efficient path planning algorithm for mobile robots based on DDPG, including the improvement of DDPG network structure, the optimization of reward function, the preprocessing of state information and the design of mixed noise. Section 4 is the simulation experiment and detailed comparison and analysis of the experimental results. Section 5 is a further summary of this paper.

## 2. Related Works

### 2.1. Deep Deterministic Policy Gradient (DDPG) Algorithm

The DDPG algorithm is based on the Actor–Critic architecture and draws on the experience replay mechanism and target network idea of DQN to solve the continuous action problem. Its network consists of the current network and target network of the Actor, and the current network and target network of the Critic. The role of the experience replay mechanism is to collect samples and sample them randomly in batches from the experience pool at each training session to reduce the correlation between samples [34]. The target network will fix the parameters in the network within a certain period of time, so as to eliminate the model oscillation caused by the same parameters between the current network and the target network [35]. The DDPG algorithm has strong fitting ability and generalization ability of deep neural network, as well as the advantage of continuous action space. Additionally, it can learn the optimal action strategy in the current state through continuous training and adjustment of neural network parameters. The method is applied to the path planning process of the mobile robot, so that the mobile robot has more continuous action output and less decision error in the process of motion. In the process of path planning, the mobile robot obtains the state S according to the surrounding environment information and its own state data, and the current network of Actor outputs the action a of the mobile robot according to S. After the mobile robot performs an action, it will obtain the reward r from the environment. According to S and a, the current network of Critic outputs the *Q* value as the evaluation of the action, and constantly adjusts its value function. The current network of Actor continuously improves the action strategy according to the *Q* value. The target network of Actor and Critic is mainly used for the subsequent update process. The structure of the DDPG algorithm is shown in Figure 1.

Initialize the current network of Actor $\mu (S|\theta_{\mu })$ and the current network of Critic $Q(S,a|\theta_{Q})$, as well as the corresponding target networks $\theta_{\mu }\prime \leftarrow \theta_{\mu }$, $\theta_{Q}\prime \leftarrow \theta_{Q}$ and experience pool D.Input the current state $S_{t}$ of the mobile robot into the current network of Actor to obtain action $a_{t}$, receive the reward r by performing the action, and obtain the next new state $S_{t+1}$ of the next step.Put $S_{t},a_{t},r,S_{t+1}$ in experience pool D. When the number of samples in the experience pool reaches a certain number, *N* samples will be randomly sampled from the experience pool D for network training.Calculate the current network loss function of Critic according to Formula (1),
(1)$$L(\theta_{Q})=\frac{1}{N}\sum_{i=1}^{N}{\left(Y_{i}-Q(S_{i},a_{i}|\theta_{Q})\right)}^{2}$$
where $Y_{i}=r_{i}+\gamma Q\prime (S_{i+1},\mu \prime (S_{i+1}|\theta_{\mu }\prime )|\theta_{Q}\prime )$ is the target value, γ is the discount coefficient, and i is the sample number of the sample.Update the current network parameters of the Actor according to Formula (2),
(2)$$\nabla_{\theta_{\mu }}J=\frac{1}{N}\sum_{i}^{N}\nabla a_{i}Q(S_{i},a_{i}|\theta_{Q})\nabla \theta_{\mu }\mu (S_{i}|\theta_{\mu })$$
where ∇J is the gradient.According to Formula (3), the target network parameter $\theta_{\mu }\prime$ of Actor and the target network parameter $\theta_{Q}\prime$ of Critic are updated using the soft update strategy,
(3)$$\{\begin{array}{c} \theta_{Q}\prime \leftarrow \tau \theta_{Q}+(1-\tau )\theta_{Q}\prime \\ \theta_{\mu }\prime \leftarrow \tau \theta_{\mu }+(1-\tau )\theta_{\mu }\prime \end{array}$$
where τ is the constant coefficient, which is used to adjust the soft update factor.Repeat the above steps until DDPG algorithm training is complete.

As one of the mainstream algorithms of deep reinforcement learning, the DDPG algorithm is widely used in mobile robot path planning. Since the algorithm adopts continuous state space and action space, it is especially suitable for the actual motion process of mobile robots, showing great potential in complex environments.

### 2.2. Long Short-Term Memory

Long Short-Term Memory (LSTM) is a special kind of Recurrent Neural Network (RNN). On the basis of fully connected neural networks, RNN adds the sequential relationship before and after and endows the network with the ability of “memory” [36]. The output of LSTM at the current moment should take into account not only the input at the current moment, but also the previous information. However, too much “memory” will also increase the computing burden of the network. LSTM introduces three gating mechanisms on the basis of RNN, namely forget gate, input gate and output gate. Through the gating mechanisms, the information at every moment is judged and adjusted in a timely fashion and updated to determine the retention degree of input information [37,38], thus reducing the burden of network computing. The structure of LSTM is shown in Figure 2.

LSTM can make full use of previous information and is suitable for processing and predicting applications with long time sequences. Mobile robot path planning is a typical long-sequence decision-making problem. In reference [31], when the DDPG algorithm is used to carry out path planning for mobile robots, each layer in the network structure adopted is a fully connected layer. It can navigate the mobile robot to the desired target position without colliding with any obstacles. However, due to the limited range of environments that mobile robots can observe and the lack of previous “memory”, path planning can only rely on the current state of the mobile robot, resulting in the planned path being too tortuous, which seriously affects the efficiency of the robot. In this paper, LSTM is introduced into the DDPG network structure proposed in reference [31], and the path planning is carried out by making comprehensive use of the past and current states of mobile robots.

## 3. The Proposed Path Planning

In order to solve the shortcomings of traditional DDPG algorithms in path planning, LSTM is introduced to optimize the structure of DDPG network, and the reward function is redesigned to speed up network training. Then, the states of the mobile robot are preprocessed by different normalization methods, such as the input of DDPG network model, and the mixed noise composed of Gaussian noise and OU noise is added to the actions output from the network model to improve the exploratory nature of the mobile robot.

### 3.1. Introduction of LSTM

When using the DDPG algorithm for path planning, the action of the mobile robot can only be determined by the current state of the robot, which will easily lead to confusion in the exploration trajectory. On the basis of reference [31], we take advantage of LSTM’s ability to memorize past states of mobile robots and introduce it into the learning process of mobile robots. To be specific, in the Actor network, the first fully connected layer is replaced by the LSTM network. When the Actor network receives the input state of the mobile robot, it is processed by the LSTM first, then processed by the two fully connected layers, and finally outputs the actions of the robot. In the Critic network, we replace the fully connected layer that processes the state with an LSTM network. When the Critic network receives the input state and action of mobile robot, the state is processed by the LSTM network, and the action is processed by the fully connected layer. The results of the above two layers are processed by a fully connected layer, and then the *Q* value is output. By comprehensively considering the current state and past state of the mobile robot, the action output by Actor network can be evaluated more accurately. In this way, the actions of the robot are controlled not only by the current state of the robot, but also by the previous state, so that the actions of the robot have time correlation, which can effectively avoid the planned path being too tortuous. The LSTM-DDPG network structure designed in our paper is shown in Figure 3.

In the Actor network and Critic network of DDPG, the target network has the same structure as the current network, and both of them adopt the structure of LSTM and fully connected layer. In the Actor network, the first layer is LSTM, the second layer is the fully connected layer with 400 nodes, and the third layer is the fully connected layer with 300 nodes. ReLU is used as the activation function of the fully connected layer, and Batch Norm layer is added after each layer to ensure the stability of the algorithm. In the Critic network, states are input to the LSTM layer, actions are input to the fully connected layer with 400 nodes, and then both of them are processed through the fully connected layer with 300 nodes. ReLU is also used as the activation function of the fully connected layer.

### 3.2. Design of the Reward Function

The reward function is a benchmark to evaluate the action taken by the mobile robot and plays a guiding role in the whole learning process. The design of the reward function should not only consider that the mobile robot can reach the target point through the optimal path, but also consider the safety of the mobile robot. After the current network of Actor outputs the action according to the robot state, the state should be updated according to the execution result of the action, and the reward value is calculated. If the mobile robot reaches the target point, the maximum positive reward will be given. If the mobile robot encounters an obstacle during its movement, it should be punished. If the mobile robot neither encounters the obstacle nor reaches the target point, the reward value should be calculated according to the distance between the mobile robot and the starting point and the target point, so that the mobile robot can keep approaching the target point. In other words, every action of the mobile robot should receive timely feedback, so as to speed up the convergence of algorithm. To achieve this goal, the reward function designed in this paper is shown in Formula (4),
(4)$$\mathrm{reward}=\{\begin{array}{cc} C_{1} & \;\mathrm{Reach}\;\mathrm{target}\;\mathrm{point}\\ C_{2} & \;\mathrm{Hit}\;\mathrm{an}\;\mathrm{obstacle}\\ -rel\_dis+ori\_dis & \;\mathrm{Other} \end{array},$$
where $C_{1}$ is a positive constant, $C_{2}$ is a negative constant, rel_dis is the distance from the mobile robot to the target point, and ori_dis is the distance from the mobile robot to the starting point. In this paper, $C_{1}$ and $C_{2}$ are set to 150 and −100, respectively.

### 3.3. State Normalization of Mobile Robot

The state space is the feedback of the whole environment of the mobile robot and is the basis for the mobile robot to select actions. The mobile robot mentioned in this paper interacts with the environment through the laser sensor. The detection range of the laser sensor is from −90 degrees to 90 degrees straight ahead, and the detection distance is at least 0.2 m. If the distance between the mobile robot and the obstacle is less than 0.2 m, it is considered to have collided with the obstacle. The data detected by the laser sensor contain 10 dimensions, as shown in Formula (5). The detection range of the laser sensor is shown in Figure 4.
(5)scan_range=[scan1,scan2,scan3,scan4,scan5,scan6,scan7,scan8,scan9,scan10],
where scan_range is the detection range of the laser sensor installed on the mobile robot, and scani is the data detected in the *i*th orientation.

In order to complete the path planning of the mobile robot, it is necessary to know whether the mobile robot will encounter obstacles, and also other states of the mobile robot, such as the action of the mobile robot at the previous time step (including linear velocity and angular velocity), the relative distance and angle between the mobile robot and the target point, the yaw angle, and the difference angle between the mobile robot and the target point, etc. Among them, the difference angle is shown in Formula (6). In order to improve the learning efficiency of the mobile robot, the states of the robot are preprocessed in different normalization methods, as shown in Formulas (7)–(10), respectively.
(6)diff_angle=|rel_eheta−yaw|,
where diff_angle is the difference angle between the mobile robot and the target point, rel_eheta is the relative angle between the mobile robot and the target point, and yaw is the yaw angle of the mobile robot.
(7)rel_dis=rel_dis/diagonal_dis,
where rel_dis is the relative distance between the mobile robot and the target point, and diagonal_dis is the diagonal length of the simulation map.
(8)rel_theta=rel_theta/360,
where rel_eheta is the relative angle between the mobile robot and the target point.
(9)yaw=yaw/360,
where yaw is the yaw angle of the mobile robot.
(10)diff_angle=diff_angle/180,
where diff_angle is the difference angle between the mobile robot and the target point.

To sum up, in the DDPG algorithm, the state space of the mobile robot is set as 16-dimensional data and defined as the input of the neural network. The normalized state $S_{t}$ can be defined as:(11)$$S_{t}=\left[a_{t-1},rel\_dis,rel\_theta,yaw,diff\_angle,scan\_range\right].$$

Considering the motion smoothness of mobile robot and the continuity of output actions, the output of the DDPG network model proposed in this paper is continuous linear velocity and angular velocity to guide the movement of the mobile robot. Since the limits of angular velocity and linear velocity should not be too large in the simulation environment, the maximum angular velocity is set to 0.5 rad/s and the maximum linear velocity is set to 0.25 m/s. The output action of the model is shown as Formula (12),
(12)$$a_{t}=[v_{t},\varphi_{t}],$$
where $a_{t}$ is the action of mobile robot at time t, while $v_{t}$ and $\varphi_{t}$ are linear velocity and angular velocity, respectively.

### 3.4. Mixed Noise Design

DDPG adopts deterministic policy with poor exploration of the environment. In order to increase the randomness of the learning process, DDPG will add a certain amount of noise to the output actions to improve the exploration ability of the mobile robot. At present, Gaussian noise and OU noise are commonly used in DDPG. Gaussian noise produces irrelevant exploration in a time sequence; that is, the selection of the front and rear actions are independent. The OU noise is a random process, and its calculation formula is shown in Formula (13). It can produce time-sequence-related exploration; that is, the action of the next step will be affected by the action of the previous step. Different from Gaussian noise, OU noise does not make the actions of two adjacent steps of the mobile robot very different, but makes the mobile robot explore near the mean of action sampling. Although this allows the mobile robot to continuously explore in one direction, it will increase the movement time of the robot when the action taken is not optimal in the current view.
(13)$$N_{\mathrm{OU}}(d_{a_{t}})=\theta (\bar{a}-a_{t})dt+\delta dW_{t},$$
where θ is the learning rate of the random process, $a_{t}$ is the action at time t, $\bar{a}$ is the average value of the action sampling data, δ is the random weight of OU, and $W_{t}$ is the Wiener process.

In order to optimize the DDPG exploration policy and improve the exploration efficiency of the robot, we combine Gaussian noise and OU noise to form mixed noise. The Actor network output action $a_{t}$ based on the mixed noise is shown in Formula (14),
(14)$$a_{t}\sim N_{\mathrm{Gaussian}}(a_{t}+N_{\mathrm{OU}}(d_{a_{t}}),\mathrm{var}),$$
where var is the Gaussian variance to ensure that the mobile robot has uniform and stable detection ability in each episode. At the same time, with the progress of the training process, the mobile robot begins to adapt to the task scene, which requires the exploration rate to be gradually reduced, as shown in Formula (15),
(15)var=var×0.9999.

This paper proposes a path planning algorithm for a mobile robot based on improved DDPG, which can solve the problems of a long training time and slow convergence of traditional path planning model. The algorithm flow is shown in Figure 5.

## 4. Experimental Results and Analysis

### 4.1. Environment Construction of Simulation Experiment

ROS is selected as the simulation experimental platform in this paper, Python and TensorFlow frameworks are used to realize the proposed algorithm, and Gazebo7 is used to establish the simulation environment, as shown in Figure 6. The black dot is the mobile robot, the blue part is the detection range of the laser sensor, and the gray part is the obstacle. Figure 6a shows the established square-shaped simulation environment without obstacles, which is mainly used to train mobile robots to realize path planning in a limited space. Figure 6b adds large obstacles to the environment of Figure 6a to train the mobile robot to realize path planning in an environment with obstacles. Figure 6c adds random small obstacles to the environment of Figure 6a, which is mainly used to test the training effect of the mobile robot in the above two environments.

We test the proposed path planning algorithm from three aspects of convergence speed, training time and success rate and analyze the experimental results in detail. The convergence speed and training time are used as evaluation criteria of the training efficiency of the algorithm, and the success rate is used to verify the effectiveness of the algorithm. In the training process of the algorithm, the convergence speed and training time can reflect how many episodes are needed to obtain the optimal solution. The faster the convergence speed is, the shorter the training time is, which means the higher the training efficiency is. The success rate refers to the percentage of mobile robots that can successfully reach the target point from the starting point according to the path planning algorithm adopted. The higher the success rate is, the better the performance of the algorithm.

### 4.2. Effect Analysis of Our Algorithm

In order to verify the performance of our algorithm, the DDPG algorithm proposed in reference [31], our algorithm with an improved network structure (LSTM-DDPG) and our algorithm after adding mixed noise further (MN-LSTM-DDPG) are all trained in the simulation environment, respectively.

Firstly, 2000 episodes of training are conducted in the simulation Figure 6a, and the reward value of the mobile robot is recorded after each episode of training, as shown in Figure 7. The results in Figure 7a show that in an environment without obstacles, the reward value of the algorithm proposed in [31] gradually tends to be stable with the increase in training episodes, but it still fails to converge after 2000 episodes of training. Moreover, the reward value fluctuates greatly in the first 800 episodes and is mostly negative, indicating that the mobile robot is learning how to approach the target point but fails many times. After 800 episodes, the reward value gradually tends to be positive, indicating that the mobile robot can reach the target point through training, but it still collides with obstacles. Figure 7b is the training result after LSTM is introduced into the network of the algorithm proposed in [31]. As can be seen, with the increase in training time, the reward value obtained by the mobile robot gradually increases from a negative value to a positive value, and finally tends to be stable, but the convergence rate is slow. After LSTM is introduced into the algorithm, the mixed noise composed of Gaussian noise and OU noise is further introduced into the output results of the network, and the reward function is optimized. Figure 7c shows the training result of the algorithm. As can be seen, with the increase in training time, the reward value obtained by the mobile robot gradually increases and finally tends to be stable. Compared with Figure 7b, the convergence speed is significantly faster.

Figure 8 shows the average rewards returned every 10,000 steps by the algorithm proposed in reference [31] and LSTM-DDPG and MN-LSTM-DDPG proposed in this paper in the path planning training in simulation Figure 6a. The blue line represents the algorithm proposed in [31], the green line represents the LSTM-DDPG algorithm, and the red line represents the MN-LSTM-DDPG algorithm. As can be seen, MN-LSTM-DDPG has the fastest convergence speed in the path planning of mobile robots, requiring only 120,000 steps, while the algorithm proposed in [31] requires 200,000 steps to converge, and the convergence is not stable.

Table 1 compares the training time and the number of training steps of the three algorithms mentioned above. As can be seen from the table, the training time of the path planning of the algorithm proposed in [31] is 28.03 h, while the training time of MN-LSTM-DDPG is only 22.75 h, which is 18.8% shorter. In terms of the number of training steps, the number of training steps of the algorithm proposed in [31] is 484,231, while the number of training steps of MN-LSTM-DDPG is only 417,701, which significantly improves the convergence speed.

In order to verify the success rate and generalization ability of the model trained in simulation Figure 6a, 200 tests are performed on the three algorithms in simulation Figure 6a and simulation Figure 6c, respectively. Figure 9 and Figure 10 show the movement process of the mobile robot in these two environments when MN-LSTM-DDPG is used for path planning. The black dot is the mobile robot, the blue part is the detection range of the laser sensor, the gray part is the obstacle, and the green circle is the final target point. As can be seen, the mobile robot can avoid obstacles from the starting point and reach the target point accurately with the optimal path.

Table 2 and Table 3 record the test results in simulation Figure 6a and simulation Figure 6c, respectively. It can be seen from Table 2 that the path planning success rate of the algorithm proposed in [31] is only 86%, while the success rate of the MN-LSTM-DDPG algorithm proposed in this paper can reach 100%. In terms of time, compared with the algorithm proposed in [31], the path planning time of the MN-LSTM-DDPG algorithm proposed in this paper is shortened by 21.48%. However, when the model trained in simulation Figure 6a is applied to simulation Figure 6c, the testing effect of each algorithm is not ideal, as shown in Table 3. Therefore, it is necessary to train the algorithm in an environment with obstacles.

In order to verify the effect of the proposed algorithm in an obstacle environment, the algorithm proposed in [31], the LSTM-DDPG algorithm and the MN-LSTM-DDPG algorithm proposed in this paper were respectively trained for 2000 episodes in simulation Figure 6b, and the average reward returned by the mobile robot every 10,000 steps was recorded, as shown in Figure 11. As can be seen from the figure, the improved MN-LSTM-DDPG algorithm in this paper can achieve convergence after 110,000 steps of training in the path planning of mobile robots in the simulation Figure 6b However, the algorithm proposed in [31] needs 200,000 training steps to converge, and the convergence is unstable; the convergence speed is significantly slower than the algorithm proposed in this paper. Additionally, from the training results in Table 4, it can be seen that the path planning training time of the algorithm proposed in the [31] is 21.73 h, while the path planning training time of the improved MN-LSTM-DDPG algorithm in this paper is only 19.70 h, and the training speed shows a significant improvement. In terms of the number of training steps, the number of training steps of the algorithm proposed in [31] is 374,316, while the number of training steps of the improved MN-LSTM-DDPG algorithm in this paper is only 346,667, which significantly improves the convergence speed of the algorithm.

After training the model in an environment with obstacles, 200 tests were performed in simulation Figure 6b and simulation Figure 6c, respectively, to verify the generalization ability of the model. The test process is shown in Figure 12 and Figure 13. The two figures respectively show the process of mobile robot avoiding obstacles from the starting point to reach the target range.

Table 5 and Table 6 record the test results in simulation Figure 6b and simulation Figure 6c, respectively. As can be seen from Table 5, in terms of success rate, the success rate of the algorithm proposed in [31] is 76%, while the success rate of the MN-LSTM-DDPG algorithm proposed in this paper can reach 87%, which is significantly higher than that in reference [31]. In terms of time, the test time of the algorithm in this paper is also 17.96% faster than the algorithm in [31]. It can be seen from Table 6 that the test effect of each algorithm in simulation Figure 6c is better than that in simulation Figure 6b. This is because when the obstacle is too large, the number of steps that the mobile robot needs to move increases. Since the maximum number of steps of the mobile robot is limited in the simulation environment, when the maximum number of steps has not reached the target point, it is regarded as a failure.

### 4.3. Comparison and Analysis with Other Algorithms

In order to fully evaluate the performance of the proposed algorithm, experiments are conducted to compare the proposed algorithm with those in references [32,33]. In the simulation Figure 6a, each algorithm is trained for 2000 episodes, and the average reward returned by the mobile robot every 10,000 steps is recorded. The results are shown in Figure 14. As can be seen from the figure, in the barrier-free environment, when the algorithm proposed in [32] carries out path planning training, it tends to converge at about 110,000 steps, but the reward value after convergence still shows a downward trend, and the training effect is not good. The algorithm proposed in [33] tends to stabilize after 210,000 steps of training. However, the algorithm proposed in this paper can converge and become stable after 120,000 steps of training, and the training effect is the best. Table 7 records the comparison of training time and the number of steps of each algorithm. As can be seen from the table, in terms of training time, the training time of the algorithm proposed in [32] is 39.63 h. The training time of the algorithm proposed in [33] is 24.67 h. However, the training time of the algorithm proposed in this paper is only 22.75 h, which is significantly faster than the algorithm proposed in [32] and better than the algorithm proposed in [33]. In terms of the number of training steps, the training steps of the algorithm proposed in references [32,33] are 395,344 and 446,596, respectively, while the training steps of the algorithm proposed in this paper are 417,701. The algorithm proposed in [32] will fall into local optimum during training, which leads to the mobile robot turning in place, and in this paper, the proposed algorithm can effectively avoid the phenomenon. Compared with reference [32], this paper proposed that although the algorithm steps of training increased, the training time is significantly reduced, and the training speed is still faster than the algorithm in [32]. However, the algorithm proposed in this paper can effectively avoid this phenomenon. Compared with the algorithm proposed in [32], although the number of training steps is increased, the training time is significantly reduced, and the training speed is still faster than the algorithm proposed in [32]. This indicates that the training effect of the algorithm proposed in this paper is better than the algorithm proposed in the references [32,33] in a barrier-free environment.

To verify the success rate and generalization ability of the model trained in simulation Figure 6a, 200 tests were conducted each in simulation Figure 6a and simulation Figure 6c, and the results are shown in Table 8 and Table 9. It can be seen from Table 8 that the path planning success rate of the algorithm proposed in [32] is 87.5%, and that of the algorithm proposed in [33] is 90%. However, the path planning success rate of the algorithm proposed in this paper can reach 100%, which is significantly higher than that in references [32,33]. In terms of time, the algorithm proposed in this paper takes the same time as the algorithm proposed in [33], which is significantly shorter than that in [32]. However, it can be seen from the test results in Table 9 that the model trained by each algorithm in a barrier-free environment is not ideal when tested in an obstacle environment, but the algorithm proposed in this paper still performs better than the other two algorithms.

In order to verify the advantages of the algorithm in this paper in an environment with obstacles, the algorithm in this paper and the algorithm proposed in references [32,33] were trained for 2000 episodes in simulation Figure 6b. The average rewards returned by the mobile robot every 10,000 steps were recorded, and the results are shown in Figure 15. As can be seen from the figure, when the algorithm proposed in [32] performs path planning in simulation Figure 6b, it needs 240,000 steps of training to converge. The algorithm proposed in [33] needs 140,000 training steps to become stable, while the algorithm proposed in this paper can achieve convergence after 110,000 training steps, and the convergence effect is obviously better than that in references [32,33]. It can also be seen from Table 10 that the training time and number of training steps taken by the algorithm proposed in this paper are significantly less than those in references [32,33].

In order to verify the success rate of the trained model in the obstacle environment, each algorithm was tested 200 times each in simulation Figure 6b and simulation Figure 6c, and the results are shown in Table 11 and Table 12. As can be seen from the test results in Table 11, in terms of success rate, the algorithm proposed in [32] is 82.5%, the algorithm proposed in [33] is 81%, and the algorithm proposed in this paper can reach 87%, which is significantly higher than the algorithm proposed in references [32,33]. In terms of test time, the test time of the algorithm proposed in this paper is longer than that proposed in [33]. In the testing process, since the starting and ending points of the mobile robot are randomly selected, their relative positions will have a certain influence on the testing time. It can be seen from Table 12, in simulation Figure 6c, the success rate of the algorithm proposed in this paper can reach 90.5%, higher than the algorithm proposed in references [32,33], which indicates that the algorithm proposed in this paper has obvious advantages in path planning in an environment with obstacles.

## 5. Conclusions

Since LSTM has the ability of “memory”, this paper uses LSTM to optimize the DDPG network structure. By designing mixed noise and more reasonable reward function, mobile robot path planning models can be rapidly trained. This effectively improves the exploration efficiency of a mobile robot in a complex environment and ensures that the mobile robot can reach the target point in a shorter time and by a better path. However, the algorithm in this paper only considers static obstacles in the environment, and dynamic obstacles are also important factors to be considered in many application scenarios. How to effectively avoid the impact of dynamic obstacles on path planning is important research content for the future.

## Figures and Tables

**Figure 1 sensors-22-03579-f001:**
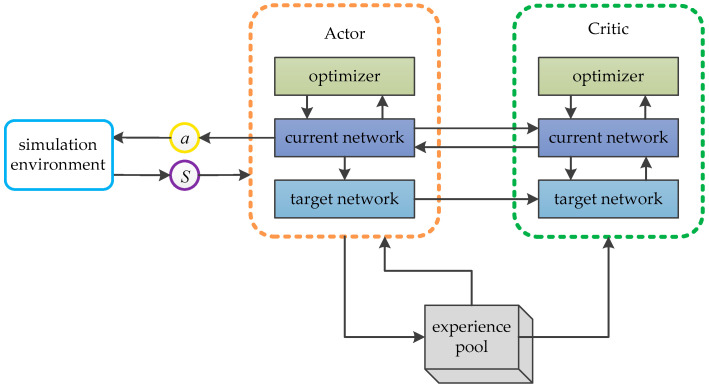
The structure of DDPG algorithm.

**Figure 2 sensors-22-03579-f002:**
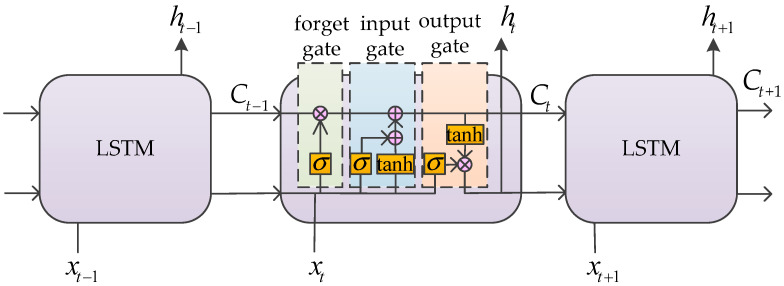
The structure of LSTM.

**Figure 3 sensors-22-03579-f003:**
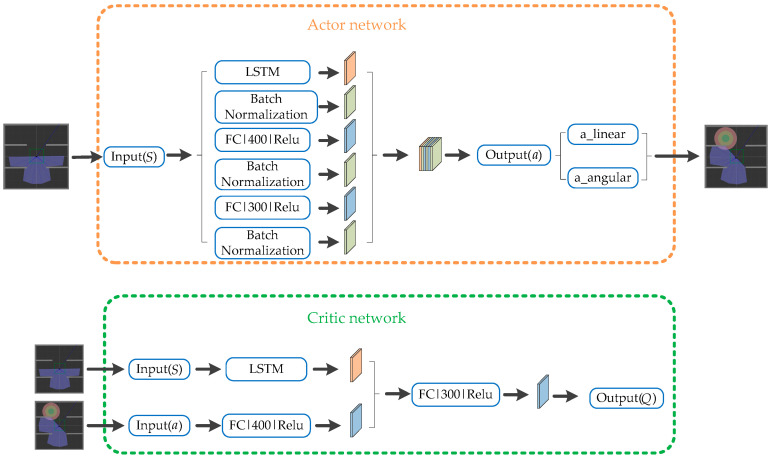
Our designed LSTM-DDPG network structure.

**Figure 4 sensors-22-03579-f004:**
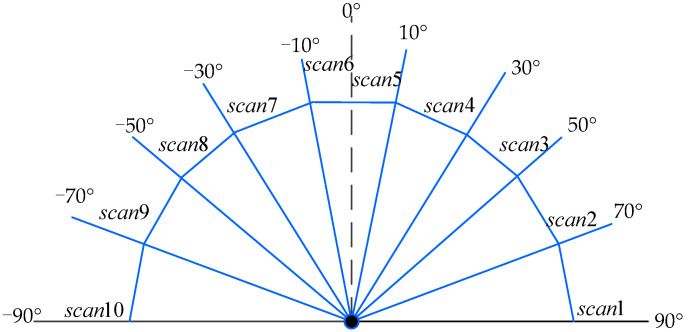
The detection range of laser sensor on mobile robot.

**Figure 5 sensors-22-03579-f005:**
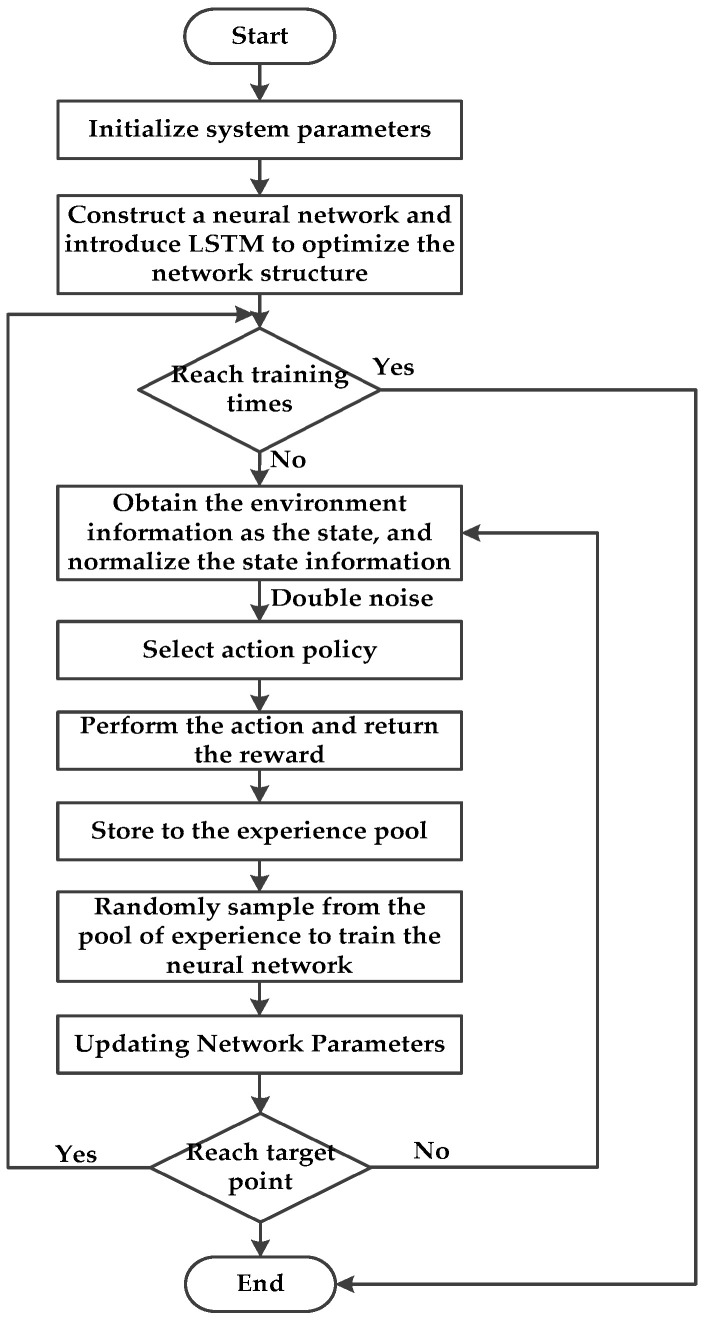
Flow chart of the proposed path planning algorithm.

**Figure 6 sensors-22-03579-f006:**
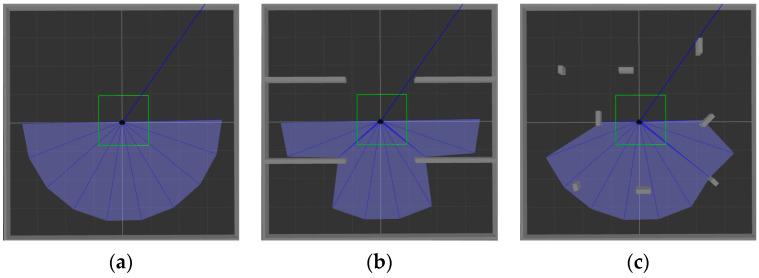
Schematic diagram of the simulation environment established by Gazebo7. (**a**) There are no obstacles in the environment; (**b**) there are large obstacles in the environment; (**c**) there are randomly generated small obstacles in the environment.

**Figure 7 sensors-22-03579-f007:**
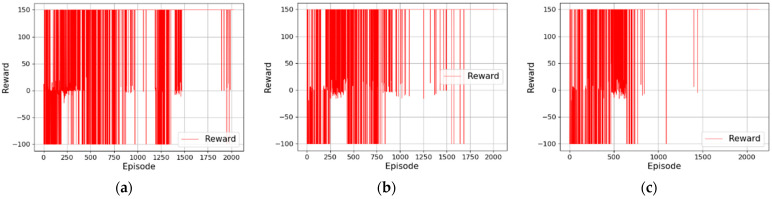
Changes in reward value of the improved algorithms. (**a**) Changes in reward value of the algorithm proposed in [31]; (**b**) changes in reward value of LSTM-DDPG, (**c**) changes in reward value of MN-LSTM-DDPG.

**Figure 8 sensors-22-03579-f008:**
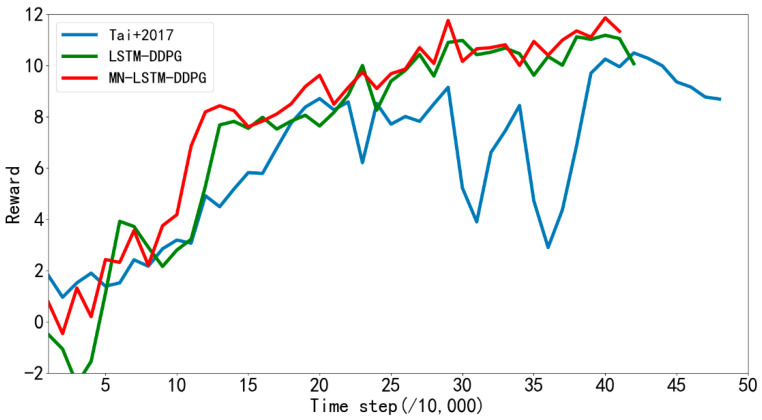
The average rewards returned by reference [31] and the improved algorithm every 10,000 steps during training in simulation Figure 6a.

**Figure 9 sensors-22-03579-f009:**
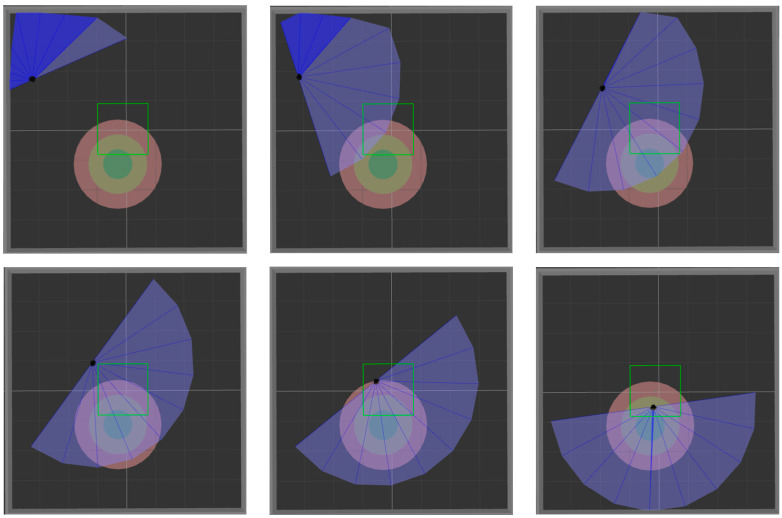
The movement process of mobile robot in simulation Figure 6a.

**Figure 10 sensors-22-03579-f010:**
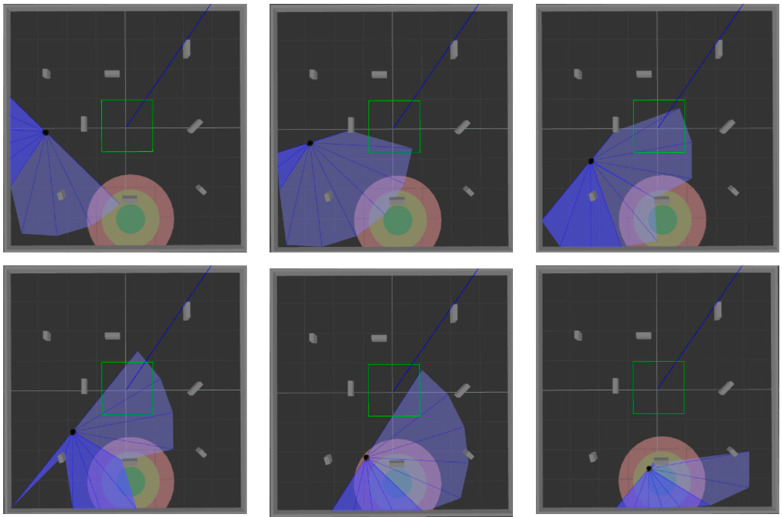
The movement process of mobile robot in simulation Figure 6c.

**Figure 11 sensors-22-03579-f011:**
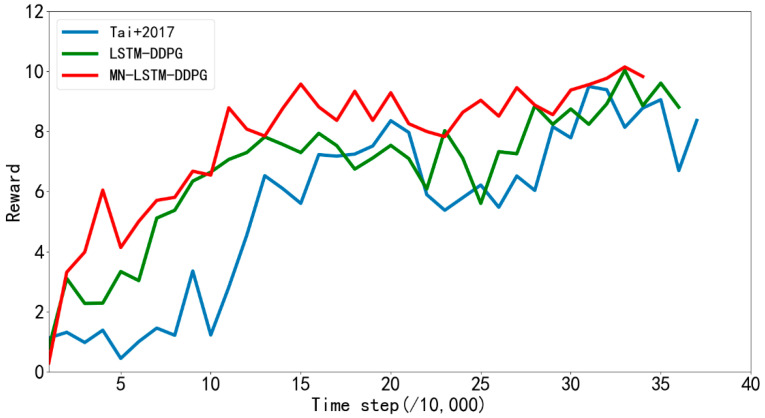
Average reward returned by reference [31] and the improved algorithm every 10,000 steps during training in simulation Figure 6b.

**Figure 12 sensors-22-03579-f012:**
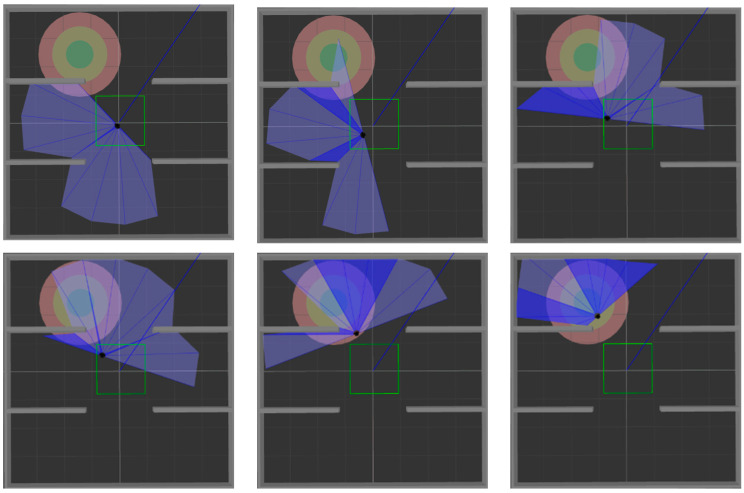
The motion process of the mobile robot in simulation Figure 6b.

**Figure 13 sensors-22-03579-f013:**
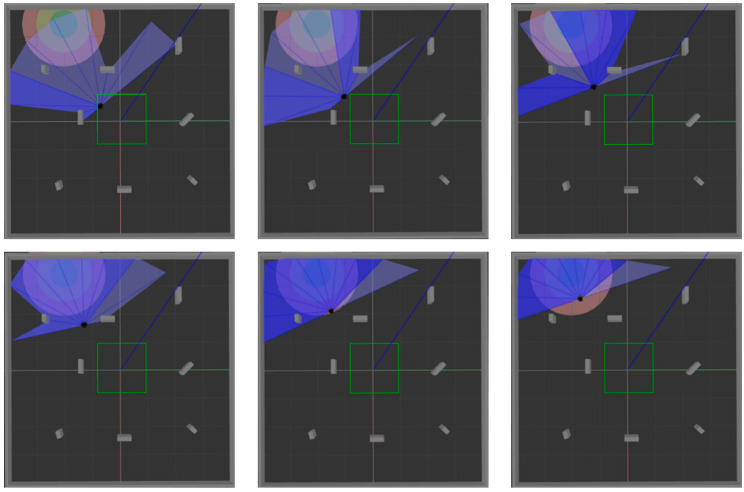
The motion process of the mobile robot in simulation Figure 6c.

**Figure 14 sensors-22-03579-f014:**
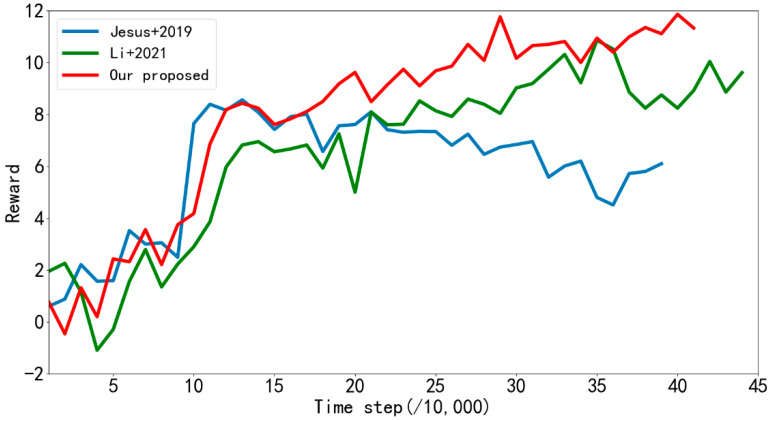
Average reward returned by reference [32,33] and our proposed algorithm every 10,000 steps during training in simulation Figure 6a.

**Figure 15 sensors-22-03579-f015:**
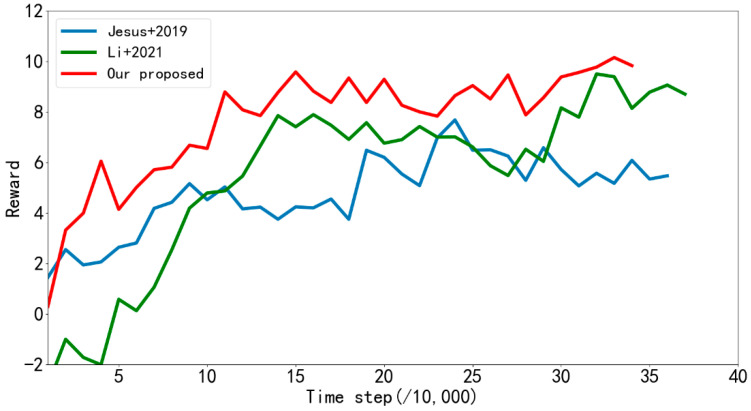
Average reward returned by references [32,33] and our proposed algorithm every 10,000 steps during training in simulation Figure 6b.

**Table 1 sensors-22-03579-t001:** Comparison of training time and steps of the improved algorithm in simulation Figure 6a.

Algorithm	Training Time (h)	Training Steps (Step)
Algorithm in [31]	28.03	484,231
LSTM-DDPG	23.68	427,956
MN-LSTM-DDPG	22.75	417,701

**Table 2 sensors-22-03579-t002:** Comparison of test results of the improved algorithm in simulation Figure 6a.

Algorithm	Success Rate (100%)	Testing Time (h)
Algorithm in [31]	172/200 = 0.86	1.35
LSTM-DDPG	194/200 = 0.97	1.21
MN-LSTM-DDPG	200/200 = 1	1.06

**Table 3 sensors-22-03579-t003:** Comparison of test results of the improved algorithm in simulation Figure 6c.

Algorithm	Success Rate (100%)	Testing Time (h)
Algorithm in [31]	108/200 = 0.54	1.88
LSTM-DDPG	133/200 = 0.665	1.71
MN-LSTM-DDPG	146/200 = 0.73	1.69

**Table 4 sensors-22-03579-t004:** Comparison of training time and steps of the improved algorithm in simulation Figure 6b.

Algorithm	Training Time (h)	Training Steps (Step)
Algorithm in [31]	21.73	374,316
LSTM-DDPG	21.20	365,164
MN-LSTM-DDPG	19.70	346,667

**Table 5 sensors-22-03579-t005:** Comparison of test results of the improved algorithm in simulation Figure 6b.

Algorithm	Success Rate (100%)	Testing Time (h)
Algorithm in [31]	152/200 = 0.76	2.06
LSTM-DDPG	165/200 = 0.825	1.55
MN-LSTM-DDPG	174/200 = 0.87	1.69

**Table 6 sensors-22-03579-t006:** Comparison of test results of the improved algorithm in simulation Figure 6c.

Algorithm	Success Rate (100%)	Testing Time (h)
Algorithm in [31]	168/200 = 0.84	1.53
LSTM-DDPG	172/200 = 0.86	1.32
MN-LSTM-DDPG	181/200 = 0.905	1.29

**Table 7 sensors-22-03579-t007:** Comparison of training time and steps of each algorithm in simulation Figure 6a.

Algorithm	Training Time (h)	Training Steps (Step)
Algorithm in [32]	39.63	395,344
Algorithm in [33]	24.67	446,596
Our proposed	22.75	417,701

**Table 8 sensors-22-03579-t008:** Comparison of test results of each algorithm in simulation Figure 6a.

Algorithm	Success Rate (100%)	Testing Time (h)
Algorithm in [32]	175/200 = 0.875	1.06
Algorithm in [33]	180/200 = 0.90	1.87
Our proposed	200/200 = 1	1.06

**Table 9 sensors-22-03579-t009:** Comparison of test results of each algorithm in simulation Figure 6c.

Algorithm	Success Rate (100%)	Testing Time (h)
Algorithm in [32]	123/200 = 0.615	1.66
Algorithm in [33]	135/200 = 0.675	1.23
Our proposed	146/200 = 0.73	1.69

**Table 10 sensors-22-03579-t010:** Comparison of training time and steps of each algorithm in simulation Figure 6b.

Algorithm	Training Time (h)	Training Steps (Step)
Algorithm in [32]	21.03	365,978
Algorithm in [33]	21.67	379,379
Our proposed	19.70	346,667

**Table 11 sensors-22-03579-t011:** Comparison of test results of each algorithm in simulation Figure 6b.

Algorithm	Success Rate (100%)	Testing Time (h)
Algorithm in [32]	165/200 = 0.825	1.71
Algorithm in [33]	162/200 = 0.81	1.35
Our proposed	174/200 = 0.87	1.69

**Table 12 sensors-22-03579-t012:** Comparison of test results of each algorithm in simulation Figure 6c.

Algorithm	Success Rate (100%)	Testing Time (h)
Algorithm in [32]	167/200 = 0.835	1.31
Algorithm in [33]	166/200 = 0.83	1.29
Our proposed	181/200 = 0.905	1.29

## Data Availability

Not applicable.

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
