# Peer review of "Efficient Path Planning for Mobile Robot Based on Deep Deterministic Policy Gradient"

_sensors, 2022, doi:10.3390/s22093579_

Round 1
Reviewer 1 Report
I have analysed this article which addresses the possibilities for efficient path planning for robots.
The authors discuss an interesting matter in the field and provide further explanation for DDPG. Their approach is quite satisfactory even though the field is difficult.
A correct procedure is employed by the author to determine their findings.
Regarding the Methodology, the paper analyses in full the DDPG algorithm and improves its success rate. In this way we believe that the learning process for mobile robots is enhanced. However, it could be beneficial for the paper to outline this methodology in a separate chapter.
It seems that the authors have validated all their research in laboratory and their findings are adequate.
The research details are appropriate and especially their 3D paths studies. The figures and tables are carefully crafted and they are easy to read and follow.
The paper describes the background in a satisfactory manner. Data on the performance under real conditions are sufficient
The output of the project is complete and justified and represents and advance in algorithm research.
The results section is promising in this sense. The references are detailed and complete.
The conclusions are coherent with the methodology and output found by the authors and the arguments described are fully demonstrated.
The manuscript should in my opinion, is thoroughly researched with copious examples, as we find no flaw in it, we encourage the authors to continue exploring the features selected beyond the current state.
Summary of evaluation: This article is could be accepted in its current form.

Reviewer 2 Report
This paper studies the path planning problem for mobile robots based on the Deep Deterministic Policy Gradient (DDPG). The investigated topic is interesting. However, the paper can be improved in the following directions:
- The full names of DDPG, LSTM, and OU in Abstract as well as in the title are suggested to be given.
- More relevant research works need to be covered to give general readers a better introduction to the studied topic. First, when stating that mobile robots are more and more widely used and play an important role in more and more fields, ‘Distributed Task Assignment for Multiple Robots Under Limited Communication Range’, ‘Integrated Task Assignment and Path Planning for Capacitated Multi-Agent Pickup and Delivery’, and ‘Distributed multi-vehicle task assignment in a time-invariant drift field with obstacles’ can be used to support the statement, where the papers have investigated efficient task assignment algorithms for multiple vehicles/robots to perform tasks in various scenarios such as package pick-up and delivery in the warehouse and visiting multiple dispersed target locations. Second, when introducing path planning methods, the optimal-control theory based method is also popular such as those used in ‘An integrated multi-population genetic algorithm for multi-vehicle task assignment in a drift field’ and ‘Clustering-based algorithms for multi-vehicle task assignment in a time-invariant drift field’.
- Some long sentences are suggested to be divided into shorter sentences for a better understanding of the paper as ‘However, when the environment is complex, due to too many state-actions, the Q tables to be maintained is too large, resulting in a sharp increase of memory consumption, and when the dimension is too large, it will also lead to dimension disaster [21].’ and ‘Taking the mobile robot to reach the target position in different 95 simulated environments as the task, creating a good reward function, and the DDPG algorithm gives the linear velocity and angular velocity of the mobile robot, but the training effect is not very ideal.’.
- The large blank space left at the bottom of page 3 can be fully used.
- Proper punctuation is needed at the end of each equation, and the ‘where’ after each equation should not be capitalized.
- A detailed description and the problem formulation of the studied path planning problem are needed to explain the objective function of the path planning and the motion dynamics of the robot.
- In the Experimental section, it is better to stress out what is the most important objective function of the path planning problem such that the results of the algorithms can be clearly compared and explained.
Reviewer 3 Report
The article presents an improved DDPG path planning method for a mobile robot. The authors introduced LSTM and noise in the original DDPG method in order to improve the success rate and efficiency of the original algorithm.
Observations:
- it is not clear why a mobile robot needs more than 20 hours of training to go from point A to point B even in the presence of obstacles.; the objective of the path search is not well specified; the authors probably suppose that the reader is familiar with the objectives of the DDPG method, which is not always true; in an environment with no obstacles why the robot does not take the direct path to the destination?
- on lines 345,345 and 347, the authors suggest that in an environment without obstacles the original method "fail to converge", after 2000 training episodes; it seams to me that the objective and the scenario of finding a proper path is not clearly specified; it is not clear how does the authors train the complex neural model
- the reward function used by the authors to train and evaluate the results seams to me too simple; for instance it does not take into consideration the length of the path made by the robot.
- in figures 9 and 10 the actual path of the robot is not represented; in figure 9, where no obstacles are seen it is not clear why some view or the robot is blue and some gray (gray should be for obstacles, but i do not see any)
- There are some errors in English:
- line 43: "The A* algorithm is improved on the Dijkstra algorithm"
- line 54: "The algorithm is proposed by simulating ..."
- line 61: with the environment..." try "As the environment"
- line 69 - should be reformulated; similarly 74-75
Round 2
Reviewer 2 Report
I appreciate the great efforts that the authors have made in response to my questions and concerns. I am satisfied with the refined manuscript.
Author Response
Thank you very much for your careful review of the paper and your affirmation of our work.